# Does the "Belt and Road Initiative" benefit the environment? Insight from analysis of intra-industry trade in environment goods

**Yacheng Zhou**[1‡], **Feiyu Liu**[2‡], **Weidong Huo**[3]*, **Changjiang Peng**[2]

1 Business School, Zhejiang Wanli University, Ningbo, China, 2 School of Economics, Chongqing Technology and Business University, Chongqing, China, 3 School of Finance and Trade, Liaoning University, Shenyang, China

‡ YZ and FL are co-first authors on this work.
* huoweidong@lnu.edu.cn

**Data Availability Statement:** All relevant data are within the manuscript and its Supporting information files.

**Funding:** This manuscript has received the awards (Supported by Social Science Planning Fund

## Abstract

The expansion of the Belt and Road Initiative (BRI) has raised a wide range of concerns about its environmental impact. Therefore, from the perspective of environmental impacts, this study used the two-way fixed effect staggered differences in differences (TWFE Staggered DID) method to examine the impact of the BRI on the Environment Goods (EGs) intra-industry trade (IIT) between China and other Belt and Road (B&R) countries, including a sample of 191 countries, covering the period from 2010 to 2019 for eliminating the impact of COVID-19 and the financial crisis in 2008 and 2009. Because only 135 countries signed a Memorandum of Understanding between 2010 and 2019, this study treated these B&R countries as the study group, and the other 73 countries (non-B&R countries) as the control group. This study described EGs using the 54 6-digit code Environment Goods in Harmonized Commodity Description and Coding System listed in the "APEC LIST OF ENVIRONMENT GOODS" published by the Asia-Pacific Economic Cooperation in 2012, and used the intra-industry trade index proposed by Grubel and Lloyd in 1971 to measuring dependent variable. The research results indicated that the BRI has significantly promoted bilateral EGs IIT. The mechanism test implied that, in addition to direct impacts, the BRI also has indirect impacts by boosting the energy restructuring of B&R countries. These results prove that the BRI has positive impacts on the environment. The heterogeneity test showed that there is a heterogeneous impact depending on the type of IIT, product categorization, B&R countries' income levels, and geographic environment. This study not only gives theoretical and empirical evidence of the positive environmental impacts of the BRI, but also provides practical guidance for the development of EGS IIT between China and B&R countries, thereby contributing to global carbon emissions reduction and environmental governance to some degree.

## Introduction

In 2019, global temperatures reached their second highest level in history and remain at historically high levels. Greenhouse gases, such as carbon dioxide, in the atmosphere continue to

Program of Liaoning Province, the Grant Number is L23BJL004).

**Competing interests:** The authors have declared that no competing interests exist.

rise, making it difficult to limit global temperature increases to no more than 1.5 to 2.0 degrees Celsius above pre-industrial levels. The increase in global temperatures has negative impacts on the productivity and lives of human beings. Frequent and more extreme weather, melting of polar glaciers leading to sea level rise, and acidification of seawater, affecting the survival of marine organisms, are becoming increasingly significant problems. In this context, reducing carbon emissions has become particularly necessary and urgent. After the signing of the Paris agreement at COP 21 (the 21st Conference of the Party of the United Nations Framework Convention on Climate Change) in 2015, climate change and carbon emissions reduction have attracted much more attention globally. The academic community has also paid increasing attention to this area of research. Scholars have widely discussed measures to decrease greenhouse gas emissions from different perspectives, such as urban transportation, energy transformation, technology innovations, energy efficiency, and the green development of various industries [1–4].

However, with increasing economic globalization, commodities can be transported from one region to another more easily. Although the importing country can reduce carbon emissions by importing high emission commodities, this causes carbon emissions to increase in the exporting country [5–7], such that global carbon emissions do not decrease, becoming a 'zero-sum' game. In this case, because Environment Goods (EGs), such as solar power generation equipment, can be used to exploit clean and renewable energy sources to decrease carbon emissions, promoting EGs trade development has become an agreed-upon strategy globally [8]. Solar power, wind power, hydropower, and other clean energy can generate power from renewable energy resources (solar radiation, wind, hydro, etc.) rather than by consuming non-renewable resources, so can contribute to reducing global carbon emissions from fossil fuel combustion. In recent years, the idea of a 100% renewable energy electricity system has been challenged. Scholars argue that any renewable energy system uses machine and material inputs that consume non-renewable energy resources, so the production of equipment and raw materials will increase carbon dioxide emissions. This reflects the importance of EGs Intra-Industry Trade (IIT). IIT is a form of trade that depends on the intra-industry international division of labour. It is characterized by a finer industrial division of labour, higher production efficiency, and lower resource consumption. Highly developed EGs IIT not only reflects a strong ability to cut bilateral carbon emissions and to govern the environment, but is also a sustainable trade form that can achieve a fair carbon emissions share for both partners in the trade agreement.

As the country with the highest carbon emissions at present, China is actively participating in global environmental governance and climate action. Since premier Zhou attended the first UN conference on the Human Environment in 1972, China has never been absent from any large-scale international conference on this topic. At the 75th UN General Assembly in 2020, President Xi Jinping announced that China will achieve peak carbon emissions by 2030 and achieve carbon neutrality by 2060. In addition to actively reducing carbon emissions, China has always followed green and sustainable development principles in international economic cooperation and aims to help other countries to reduce carbon emissions under the Belt and Road Initiative (BRI). In 2015, the "Vision and Action Plan on Jointly Building the Silk Road Economic Belt and the 21st Century Maritime Silk Road" (abbreviated as "Vision and Action Plan") was issued by the Ministry of Commerce of the People's Republic of China and is a specific implementation plan of the BRI that proposes jointly constructing the green silk road. In 2017, the "Guidance on Promoting Green Belt and Road", jointly issued by the Ministry of Ecology and Environment of China and three other departments, promotes green development, strengthening eco-environment protection, and jointly building a green silk road. In the same year, President Xi Jinping proposed establishing the "BRI International Green Development Coalition", which aims to support the achievement of the environmental goals of the

2030 Sustainable Development Goals (SDGs). China has also noted in the "Position Paper on China's Cooperation with the United Nations" that the country will promote the effective integration of the BRI with the SDGs and support global climate and environmental governance. From the perspective of environmental governance and green development, the BRI and the SDGs have a common purpose. On March 28, 2022, four Chinese departments jointly issued the "Opinions on Jointly Promoting Green Development of the Belt and Road", which proposed that international cooperation on ecological environment protection and climate governance along the Belt and Road (B&R) will achieve significant results by 2025, and all parties will recognize the concept of the Green Silk Road. It also promotes cooperation in green infrastructure, green energy, green transportation, green finance, green trade, and other fields, and comprehensively promotes the green development of the B&R. Furthermore, a green development pattern of the B&R will be formed by 2030. Therefore, protecting the environment while fostering economic development is the essential goal of the BRI.

Although some scholars have argued that green trade, finance, and investment, and green technology and innovation are the essential mechanisms through which the BRI can make progress in achieving the SDGs, the BRI's environmental impacts and its implications for environmental governance have been questioned in recent years [9, 10]. Considering that green trade, especially intra-industry trade, has a profound significance for the environment, this study examined how the BRI impacts the EGs IIT between China and other countries (hereafter referred to as B&R countries) that have signed a Memorandum of Understanding (MoU) regarding cooperation on the BRI, so as to study whether the BRI has positive environmental impacts. Specifically, this study described EGs using the 54 6-digit code EGs in Harmonized Commodity Description and Coding System (HS) listed in the "APEC LIST OF ENVIRON-MENT GOODS" published by the Asia-Pacific Economic Cooperation (APEC) in 2012 [11], and used this as the basis for calculating the EGs IIT index. To eliminate the impact of COVID-19 and the financial crisis in 2008 and 2009, the sample period in this study was 2010–2019. Because only 135 countries signed a MoU between 2010 and 2019 [12], this study treated these countries as the study group, and the other 73 countries (non-B&R countries) as the control group. Since the time of signing a MoU was different for each country, this study used the method of Athey and Imbens (2022) for reference, and the two-way fixed effect staggered differences in differences (TWFE Staggered DID) was used to examine the impact of the BRI on bilateral EGs IIT between China and B&R countries [13].

In addition to the variables of the EGs IIT index and policy effect dummy variable, we also added the China Outward Foreign Direct Investment (OFDI) in B&R countries, and China and the B&R countries' average trade liberalization level, income, technology, and infrastructure similarity as control variables. In addition, the fixed effects at the country level and year level were also added to eliminate the interference of other factors with the regression results. This study's baseline regression results showed that the BRI can significantly promote bilateral EGs IIT and achieve real environmental gains among B&R countries. The mechanism tests indicated that China's BRI not only directly promotes bilateral EGs IIT, but also has an indirect impact by boosting the energy restructuring of B&R countries, which further demonstrates that the BRI adheres to green and sustainable development principles and is conducive to increasing the environmental sustainability of the B&R countries.

The main marginal contributions of this study are as follows. First, previous literature has focused mainly on the impact of the BRI on trade volume and its binary marginal effects [14–17], but this study expanded the research agenda on the BRI and trade and evaluated whether the BRI makes contributions to EGs IIT. More importantly, it linked the BRI and the B&R countries' environmental governance and answered the question regarding the environmental impacts of the BRI.

Second, this study enriched the research perspective of green trade. Since joining the WTO, China's trade has proliferated. However, the extensive growth mode in trade has received widespread attention from domestic and foreign scholars due to its negative environmental impact [18–21]. Some scholars have examined CO2 emissions embodied in trade and reflected on how to develop low-carbon green trade [22–24]. Previous literature has mainly focused on the macro-level of green trade and ignored micro-level EGs IIT. This study thus focused on EGs IIT.

The content of this paper is arranged as follows. Mechanisms and Hypotheses section analyses the impact mechanism of the BRI on bilateral EGs IIT and sets out the hypotheses. Methodology section introduces the identification strategy, model setting, data sources, and data treatments. Result and Discussion section shows the empirical regression results and discussion. Policy Implications section puts forward the policy implications.

## Mechanisms and hypotheses

In 2017, the "Opinions on Promoting the Construction of Green B&R" aimed to promote the development of green trade. On March 28, 2022, the "Opinions on Jointly Promoting Green Development of B&R" also proposed vigorously developing green product trade and expanding the import and export of environmentally friendly products (or services). Yao et al. (2021) found that China's IIT of EGs with countries along the B&R has developed rapidly since the BRI was proposed [25]. It seems that there is a causal relationship between the two. In this section, this causal relationship will be discussed in detail.

### The relationship between the B&R and bilateral EGs IIT

This study describes the relationship between the B&R and bilateral EGs IIT within the framework of "New Trade Theory". Falvey (1981) argued that a country with low barriers to trade will have a high level of IIT [26], and Leamer (1988) and Harrigan (1994, 1996) proposed that market openness has a positive relationship with IIT [27–29]. In reality, market openness can play a role in promoting intra-industry trade only in an open international investment and trade environment in which scale economics and product differentiation can be formed more easily. Scale economies and product differentiation have been demonstrated as two key impact factors of intra-industry trade by Krugman (1981)'s monopoly competition model, and can be strongly conducive to the growth of intra-industry trade [30]. For the more, the formation of scale economies and product differentiation depends on a sophisticated intra-industry international division of labour to some degree, which was discussed in the study by Pomfret (1986) [31].

The BRI carries the spirit of Silk Road in terms of win–win outcomes and adheres to the principle of cooperation through extensive consultation, joint construction, and shared benefits. It has thereby been approved by many countries around the world. The BRI's main tasks are policy coordination, unimpeded trade, financial integration, facilities connectivity, and people-to-people bonds. Therefore, building an open market environment is an inevitable tendency of BRI cooperation. Green trade has always been an important part of B&R construction. In 2015, the "Vision and Actions Plan" clearly indicated that we should vigorously promote cooperation in clean and renewable energy, such as hydropower, nuclear power, wind power, and solar energy, create an upstream and downstream integrated industrial chain of energy resources, and strengthen cooperation in energy resource deep processing technology, equipment, and engineering services. EGs in the "APEC LIST OF ENVIRONMENT GOODS" released by APEC are mainly used for power generation and deep processing of renewable energy resources, such as water, wind, and solar. Therefore, it is reasonable to

deduce that the BRI is conducive to promoting the formation of a deep vertical and horizontal intra-industry international division of labour in the field of EGs, laying the foundation for EGs IIT growth. According to the "New Trade Theory", we argue that the BRI will promote B&R countries to further open their markets to China and the intra-industry international division of labour, thereby accelerating the growth of EGs IIT.

## The relationship between the BRI and energy restructuring of B&R countries

Partly in response to questions from the international community, several Chinese ministries have issued policies on "Green Belt and Road Construction" and "Green Development of the Belt and Road", so as to protect the environment of B&R countries while fostering economic development. "Green Belt and Road Construction" and "Green Development of the Belt and Road" are essential for achieving the green transition of B&R countries [32], and the channels to promote energy restructuring of B&R countries are as follows.

First, the Chinese government and its policies can positively impact the OFDI activities of China's outward enterprises [33]. In 2013, "Guidelines for Environmental Protection in Foreign Investment and Cooperation", issued by the Ministry of Commerce and Ecology and Environment of China, encouraged enterprises to protect the host countries' environment. This will increase the demand for products exploiting renewable energy resources, monitoring the environment and pollution, treatment of waste, pollutants, etc., thereby accelerating the energy restructuring of B&R countries.

Secondly, during the construction of the B&R, the vast production capacity caused by enterprises' OFDI will increase the energy demand [34]. Traditional energy will not be able to meet the dual requirements of energy supply and environmental protection, which will increase the consumption of clean and renewable energy, thereby promoting the energy restructuring of B&R countries [35]. Wu et al. (2021) found that the BRI can significantly reduce carbon emission intensity in energy-intensity industries, which is partly due to energy restructuring [36].

Finally, investment under the BRI tends to occur in the renewable energy infrastructure sector and will help many B&R countries restructure their energy mix. As can be seen in the "Big Data Analysis Report of Ecological and Environmental Protection of the BRI 2022", new contracts in the environmental and energy field mainly include projects such as photovoltaic power, wind power, hydropower, etc. We believe that the BRI, as a far-reaching infrastructure development and investment strategy [37], can boost the energy restructuring of B&R countries.

## The relationship between energy restructuring of B&R countries and bilateral EGs IIT

To face global climate change, EGs IIT will play a pivotal role, as it fosters climate governance and is conducive to the fair allocation of carbon emissions. However, EGs IIT has attracted limited attention, and research on the relationship between the energy mix and EGs IIT remains sparse.

The causal relationship between the energy mix and EGs IIT is also discussed in the framework of "New Trade Theory". Linder (1961) proposed "Demand Similarity Theory", in which he argued that residents from two countries with similar per capital income have similar preferences for products or services, which promotes the formation of IIT between two such countries [38]. Based on this theory, we can deduce that when two countries have a similar demand for EGs, it is possible to form EGs IIT between these two countries.

However, because countries that use a large amount of renewable energy resources usually attach importance to environmental protection, their demand for EGs is relatively greater. As a result, the degree of energy restructuring that is signified by the share of hydropower in total power generation is one of the appropriate factors to represent the demand level of EGs. In other words, the deeper the level of energy restructuring, the greater the demand for EG.

At present, most B&R countries are developing countries, and developing countries, especially the African countries, mainly rather use coal, oil, and natural gas rather than renewable energy, so their level of energy restructuring or demand for EGs is lower than that of China. In this case, a deeper level of energy restructuring of B&R countries or the greater the demand for EGs, the more similar the demand for EGs between these countries and China, so that bilateral EGs IIT can be promoted. This was consistent with the study of Hu and Ma (1999), who found that China's IIT is positively related to the absolute per capital income level of its partner countries [39]. The absolute per capital income level indicates the demand level to some degree. In summary, the energy restructuring of B&R countries can stimulate bilateral EGs IIT.

The above analyses were conducted from the perspective of demand. However, from the perspective of supply, the shallow level of energy restructuring in developing countries is accompanied by small scale and undeveloped technology of EGs manufacturing industries, meaning that these countries cannot participate in EGs trade on a large scale, as prosperous EGs trade also depends on a sound and developed green industrial system. This also results in the energy restructuring having a critical impact on bilateral EGs IIT.

## Hypotheses

According to the discussion above, hypothesis 1 and hypothesis 2 are proposed as follows. The relations argued in following hypotheses are shown in Fig 1.

H1: The BRI can promote bilateral EGs IIT between China and B&R countries.

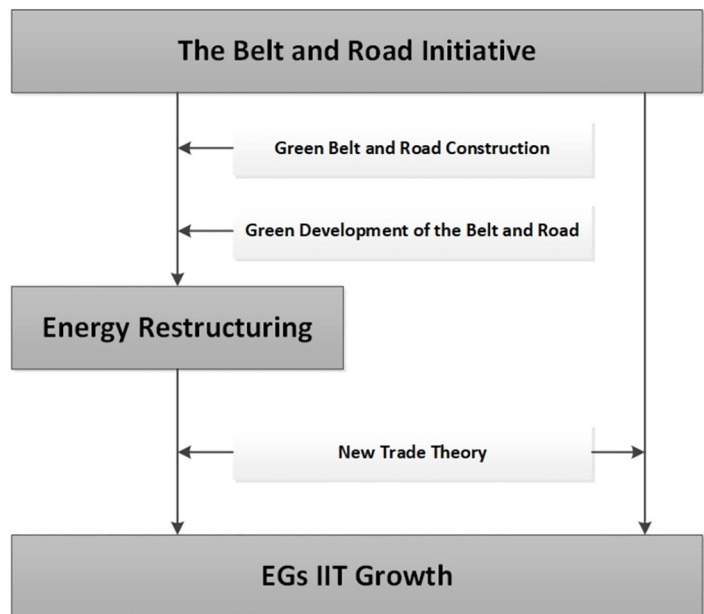

**Fig 1. Conceptual framework.**

H2: The BRI can also promote bilateral EGs IIT between China and B&R countries by boosting the energy restructuring of B&R countries.

## Methodology

### Identification strategy

One of the best and most commonly used strategies for identifying policy effects is the two-way fixed effect staggered differences in differences (TWFE Staggered DID). This study will use this method to explore the impact of the BRI on the EGs IIT between China and other B&R countries. To eliminate the interference of COVID-19 and the financial crisis in 2008 and 2009, this study will use a sample of 191 countries, covering the period from 2010 to 2019. Because only 135 countries had signed a MoU between 2010 and 2019, this study treated these countries as the study group, and the other 73 non-B&R countries as the control group. This study will set a dummy variable (*DID*) to represent the effect of the BRI on the EGs IIT. The significantly positive regression coefficient signifies that the BRI has a positive impact on the EGs IIT, otherwise, it indicates no significant impact.

### Empirical model

Drawing on Callaway and Sant'Anna (2021) [40], this study used the two-way fixed effect staggered differences in differences (TWFE Staggered DID) method to examine the impact of the BRI on bilateral EGs IIT between China and B&R countries and set up the following model:

$$IITI_{it} = \alpha_0 + \alpha_1 DID_{it} + \sum_k \beta_k X_{kit} + \gamma_i + \delta_t + \mu_{it}, \tag{1}$$

where *i* denotes the country, *t* is the year, *IITI* is the IIT index in EGs, *DID* represents the policy effect of the BRI, *X* signifies a series of control variables, $\gamma$ is the country fixed effects, $\delta$ is the year fixed effects, and $\mu$ represents the stochastic error.

### Variables specification

**Dependent variable.** The dependent variable in this study was bilateral EGs IIT index between China and B&R countries. This study defined EGs as the 54 6-digit HS EGs listed in the "APEC LIST OF ENVIRONMENT GOODS". The steps to calculate the overall IIT index of the 54 EGs were as follows. First, we matched the 6-digit HS code with the SITC Rev.4 code to obtain 14 3-digit SITC EGs. Second, we calculated the import and export trade data of the 14 3-digit SITC EGs. Third, referring to Grubel and Lloyd (1971), we used the following Eq (2) to calculate the IIT index of the 14 EGs, respectively [41]. Finally, taking the proportion of the IIT scale of each 3-digit SITC product to the IIT scale of all 14 EGs as the weight, the IIT indices of the 14 3-digit SITC EGs were added together to obtain the IIT index at the country-year level [42]. The value of the IIT index was between 0 and 1, where 0 means no IIT, and 1 means complete IIT.

$$IITI_{itk} = 1 - \frac{|x_{itk} - m_{itk}|}{|x_{itk} + m_{itk}|}, \tag{2}$$

**Independent variable.** The core independent variable in this study was the policy effect of the BRI (*DID*). *DID* was obtained by multiplying the treatment group dummy (*TREAT*) and the treatment period dummy (*PERIOD*). To exclude the impact of the 2008 financial crisis and

COVID-19, the sample period of this study was from 2010 to 2019. By the end of 2019, 135 countries had signed a MoU and were regarded as treatment group. At this time, the value of *TREAT* was 1. There were 73 countries in the control group, and the value of *TREAT* was 0. When the year of each sample in treatment group was later than or equal to the year the country in this sample signed a MoU, *PERIOD* took the value of 1. Otherwise, it took the value of 0. *PERIOD* always took the value of 0 in control group. Finally, *DID* could be calculated by multiplying *TREAT* and *PERIOD*.

**Control Variables.** **China OFDI** in B&R countries. Since the relationship between OFDI and IIT has been verified by a great number of researchers [43–48], we added China's OFDI in B&R countries as a control Variable. We assumed that the bigger the OFDI was, the more prosperous bilateral EGs IIT was.

*Income similarity.* The similarity in income level was one of the control variables in this study. According to Linder's "Theory of Preference Similarity", residents of two countries with similar income levels have more similar demand for products or services, which is the one of the motivations for forming IIT [38]. Therefore, this study considers bilateral income similarity as a control variable. We used the similarity in per capital GDP to measure bilateral income similarity between China and B&R countries. This similarity was calculated using the deformation formula of the "Balassa–Bauwens relative difference index" proposed by Balassa and Bauwens in 1987 (similarly hereinafter) [49], and the calculation formula was as follows:

$$\text{Similarity} = \ln\left[1/\left[1 + \frac{w \times \ln(w) + (1-w) \times \ln(1-w)}{\ln 2}\right]\right], \tag{3}$$

where the value of Similarity was within the range from 0 to $+\infty$. w = per capital GDP of China / (per capital GDP of China + per capital GDP of certain B&R country). There were two reasons for using this index to calculate the similarity in income level. On the one hand, when the two countries' income levels were close, w was close to 1/2, the Similarity was close to $+\infty$, but its value is actually small by taking logarithms. When the two countries' income levels was quite different, w was close to 0 or 1, which resulted in Similarity being close to 0. On the other hand, this index can eliminate the impact of measurement units on regression results.

*Bilateral average trade liberalization level.* The improvement of the trade liberalization level is convenient for bilateral trade, making it easier to form the industrial division of labour which lays the foundation for bilateral IIT growth [27–29]. As a result, bilateral trade liberalization level has an important relationship with EGs IIT between China and B&R countries. This study used the freedom to trade internationally index in the Economic Freedom of the World database released by the Fraser Institute to measure the trade liberalization level and calculate bilateral average trade liberalization level between China and B&R countries. We assumed that the trade liberalization level was positively related to bilateral EGs IIT.

*Technology similarity.* Referring to relevant literature, we added the technology level similarity between China and each B&R country as one of control variables in this study [50–52]. This study used the number of patent applications in a country to measure its technology level. We expected that the greater the similarity in technology between two countries, the more developed their bilateral EGs IIT is.

*Infrastructure similarity.* Infrastructures, especially communication and transportation infrastructure, play an important role in promoting trade growth. We introduced these two factors as control variables. We used the number of fixed broadband subscriptions (per 100 people) to measure communication infrastructure level, and railway mileage for the transportation infrastructure level. When the similarity in communication or transportation infrastructure between two countries is great, bilateral EGs IIT is more prosperous [53–55].

### Data sources and data processing

The trade data to measure bilateral EGs IIT index were from the United Nations Comtrade database. The time of signing a MoU was from the Belt and Road Portal. The per capital GDP, patent applications number, fixed broadband subscriptions, and railway mileage data were all obtained from World Bank Open Data. OFDI data were collected from the Ministry of Commerce of the People's Republic of China. The freedom to trade internationally index data were from the Economic Freedom of the World database released by the Fraser Institute. All the variables except *DID* were added by 1 and introduced to the model in a natural logarithm form.

## Results and discussion

### Parallel trend test

The premise of using the DID method is to satisfy the parallel trend assumption. Drawing on Beck, Levine, and Levkov (2010) [56], we conducted a parallel trend test, and its steps were as follows: First, this study set the variable D_1 to represent the marginal treatment effect of 1 year before the signing of a MoU. If a country signed a MoU in 2014, when year = 2013, D_1 was 1. In other years, D_1 was 0. Analogously, D_2, D_3 and D_4 were set, and D_4 was used to represent the treatment effect of 4 years or more before signing a MoU. Secondly, D represented the marginal treatment effect in the year when a MoU was signed. Thirdly, D1–D4 represented the marginal treatment effect of N (N = 1/4) years after the signing of a MoU. The control group took all marginal treatment effect variables as 0. Finally, D_3–D_1, D, and D1–D4 were introduced into the formula (1) and drew the value range of the regression coefficient of the treatment effect variable mentioned above at a 90% confidence interval, as shown in Fig 2.

It can be seen from Fig 2 that the value range of the coefficient of D_3–D_1 was not significantly different from 0, which means that the assumption of parallel trends held, and the TWFE Staggered DID could be used for causal identification.

### Baseline regression

The baseline regression results are shown in Table 1. Column (1) of Table 1 shows the *DID* regression coefficient was significantly positive without adding the country fixed effect, year

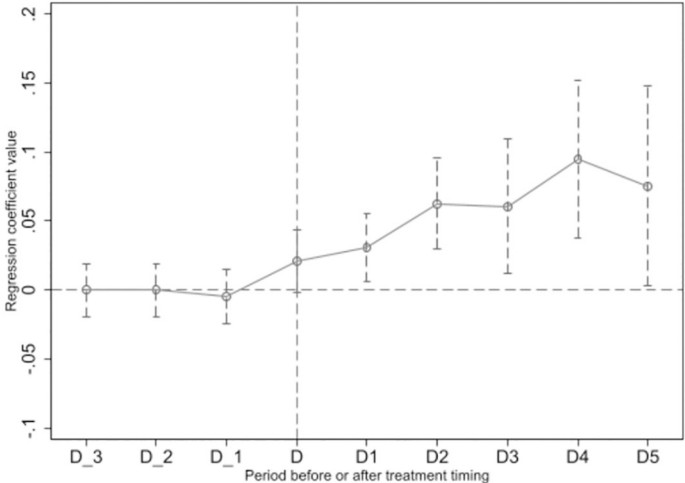

**Fig 2. Parallel trend test.**

**Table 1. Impact of the BRI on bilateral EGs IIT between China and B&R countries.**

| Variables | (1) | (2) | (3) | (4) |
|---|---|---|---|---|
| DID | 0.0443*** | 0.0343*** | 0.0333*** | 0.0306*** |
| | (0.0133) | (0.0074) | (0.0098) | (0.0097) |
| lnOFDI | | | | 0.0004 |
| | | | | (0.0032) |
| lnPCGDP | | | | 0.0068 |
| | | | | (0.0194) |
| lnTRAO | | | | 0.1721** |
| | | | | (0.0706) |
| lnTECH | | | | 0.5299*** |
| | | | | (0.1473) |
| lnCINF | | | | 0.0329*** |
| | | | | (0.0108) |
| lnTINF | | | | 0.0886* |
| | | | | (0.0453) |
| Country | no | yes | yes | yes |
| Year | no | no | yes | yes |
| Observations | 1910 | 1910 | 1910 | 1910 |
| R-squared | 0.0069 | 0.7758 | 0.7766 | 0.7806 |

Note: Parentheses show standard errors;

***, **, and * indicate statistical significance at 1%, 5%, and 10%, respectively.

fixed effect, and control variables. However, when only one independent variable was added, the regression results were unreliable. We further added not only the country fixed effect and year fixed effect, but also a series of control variables containing China's OFDI in B&R countries, the mean level of bilateral (China and certain B&R country) trade liberalization, and bilateral income, technology and infrastructure (including communication and transportation infrastructure) level similarity between China and B&R countries in column (4) of Table 1. The regression results indicated that, in the process of gradually adding all the variables from column (1) to column (4) of Table 1, the significance and impact direction of the core independent variable DID did not change, and its coefficient value did not fluctuate significantly. After adding all of the above variables, bilateral EGs IIT index increased by 3.06% due to the BRI. In a word, the BRI can significantly promote bilateral EGs IIT between China and B&R countries, which indicates that the BRI is beneficial to the environment of B&R countries.

Moreover, the regression results of the control variables were basically as expected. Specifically, according to the regression estimations of all fixed effects and control variables included in column (4) of Table 1, the mean level of bilateral trade liberalization was significantly and positively related to bilateral EGs IIT. This showed that the freer bilateral trade is, the more prosperous bilateral EGs IIT is, which is in line with economic common sense. Bilateral technology and infrastructure (including communication and transportation infrastructure) level similarity were significantly and positively correlated with bilateral EGs IIT, which indicated that when there is a smaller gap of bilateral technology or infrastructure level, bilateral EGs IIT will be more developed. This is also in line with expectations. However, China's OFDI in B&R countries had no significant relationship with bilateral EGs IIT. This may have been due to the fact that China's OFDI participates in various sectors, not only the energy infrastructure sector, which means China's OFDI had limited impacts in bilateral EGs IIT. bilateral income level

similarity was also not significantly but positively correlated with bilateral EGs IIT, which is inline with the "Theory of Preference Similarity".

## Robustness test

**Placebo test.** Referring to the practice of Topalova (2010) [57], this study advanced the treatment time by 1 to 3 years and delayed it by 1 to 3 years respectively, and the regression results are shown in Table 2. There was no significant relationship between the BRI and the EGs IIT index when the signing time of a MoU was advanced by 1 to 3 years, from column (1) to column (3) of Table 2. However, when the policy was delayed by 1 to 3 years from column (4) to column (6) of Table 2, the estimation results showed that the BRI significantly and positively promoted the EGs IIT, and its impact intensity showed an increasing trend compared to the baseline regression. This proved the robustness of the baseline regression results.

**Changing the identification strategy.** Bacon (2021) showed that the effect estimation by TWFE Staggered DID is a weighted average of all possible two-group/two-period *DID* estimators, which may influence the accuracy of TWFE Staggered DID estimation [58]. In this case, alternative causal identification methods, such as Stacked DID, Event Study, Reweighting Estimation, et al., are required. Referring to Baker et al. (2022), this study used TWFE Stacked DID to test the robustness of the baseline regression [59]. There are three steps in TWFE Stacked DID: First, the sample is divided into several subset samples according to the treatment time. Second, the treatment group and control group are selected in each subset sample. Third, all subset samples are merged, and coefficient of the core independent variable *DID* are estimated. As shown in column (2) of Table 3, the regression results indicated that the significance and impact direction of the core independent variable did not change, and the impact scale increased. This indicated that the coefficient value of the *DID* in TWFE Staggered DID estimating was underestimated. In conclusion, the estimation of the baseline regression was robust.

**Table 2. Placebo test.**

| Variables | (1) | (2) | (3) |
|---|---|---|---|
| *DID* | -0.0004 | 0.0056 | 0.0120 |
| | (0.0085) | (0.0088) | (0.0088) |
| *Control* | yes | yes | yes |
| *Country* | yes | yes | yes |
| *Year* | yes | yes | yes |
| Observations | 1910 | 1910 | 1910 |
| R-squared | 0.7793 | 0.7793 | 0.7795 |
| **Variables** | **(4)** | **(5)** | **(6)** |
| *DID* | 0.0338*** | 0.0472*** | 0.0413* |
| | (0.0114) | (0.0151) | (0.0215) |
| *Control* | yes | yes | yes |
| *Country* | yes | yes | yes |
| *Year* | yes | yes | yes |
| Observations | 1910 | 1910 | 1910 |
| R-squared | 0.7805 | 0.7808 | 0.7799 |

Note: Parentheses show standard errors;

***, **, and * indicate statistical significance at 1%, 5%, and 10%, respectively.

**Table 3. Regression results after changing the identification strategy.**

| Variables | (1) | (2) |
|---|---|---|
| *DID* | 0.0306*** | 0.0395*** |
| | (0.0097) | (0.0134) |
| *Control* | yes | yes |
| *Country* | yes | yes |
| *Year* | yes | yes |
| Observations | 1910 | 4710 |
| R-squared | 0.7806 | 0.8122 |

Note: Parentheses show standard errors;

***, **, and * indicate statistical significance at 1%, 5%, and 10%, respectively.

**Excluding the interference of expectation factors.** The signing of a MoU required a lengthy negotiation process. During the negotiation process, multinational corporations from different countries had different expectations about the possible impact of the BRI, and thus had different reactions and actions. If this expectation is not considered, the estimate results will deviate from reality. Therefore, this study added the marginal treatment effect dummy variable of 1 year before the signing of a MoU to represent the impact of expectations on the EGs IIT. The regression results are shown in column (2) of Table 4, and column (1) shows the baseline regression results. From the regression results, we found that expectations had no significant relationship with the EGs IIT, and the coefficient value of *DID* did not change significantly compared to baseline regression. Therefore, the baseline regression results were robust.

**Excluding the interference of other policy factors.** On the one hand, the list of EGs in this study was jointly signed and issued by APEC members, which may have led to the EGs IIT between China and other APEC members being different from that of China and non-APEC members. Therefore, this study excluded observations from APEC members, and the regression results are shown in column (1) of Table 5. On the other hand, because the free trade agreement (FTA) has important impact on bilateral trade, this study added dummy variable *FTA* to represent the impact of FTA on the EGs IIT. When the year in each sample is later than or equal to the time when B&R countries signed the FTA with China, the dummy variable *FTA* will take the value of 1. Otherwise, it take the value of 0. The regression results are shown

**Table 4. Regression results under controlling the expected factors.**

| Variables | (1) | (2) |
|---|---|---|
| *DID* | 0.0306*** | 0.0279*** |
| | (0.0097) | (0.0104) |
| *EXPECT* | | -0.0071 |
| | | (0.0106) |
| *Control* | yes | yes |
| *Country* | yes | yes |
| *Year* | yes | yes |
| Observations | 1910 | 1910 |
| R-squared | 0.7806 | 0.7806 |

Note: Parentheses show standard errors;

***, **, and * indicate statistical significance at 1%, 5%, and 10%, respectively.

**Table 5. Regression results when considering the interference of other policy.**

| Variables | (1) | (2) |
|---|---|---|
| DID | 0.0318*** | 0.0299*** |
|  | (0.0104) | (0.0097) |
| FTA |  | -0.0585** |
|  |  | (0.0258) |
| Control | yes | yes |
| Country | yes | yes |
| Year | yes | yes |
| Observations | 1730 | 1910 |
| R-squared | 0.7500 | 0.7810 |

Note: Parentheses show standard errors;

***, **, and * indicate statistical significance at 1%, 5%, and 10%, respectively.

in column (2) of Table 5. The regression results imply that the significance and influencing direction of the *DID*, when excluding APEC members or considering the impact of FTA on the EGs IIT, were in line with the baseline regression, and the influencing scale was relatively stable. Thus, the baseline regression was robust.

## Mechanism test

As discussed in hypothesis 1 and hypothesis 2, the BRI can directly promote bilateral EGs IIT, and can also have an indirectly positive impact by promoting energy restructuring of B&R countries. This section will test the above hypotheses. This study selected the logarithm of the share of hydropower in total electricity production to represent the level of energy restructuring of B&R countries as the intermediary variable and used the mediation effect model to test this impact mechanism. From this, we set up the following models [60]:

$$\ln IITI_{it} = \alpha_0 + \alpha_1 DID_{it} + \sum_k \beta_k \ln X_{kit} + \gamma_i + \delta_t + \mu_{it}, \tag{4}$$

$$\ln ER_{it} = \alpha_0 + \alpha_2 DID_{it} + \sum_k \beta_k \ln W_{kit} + \gamma_i + \delta_t + \mu_{it}, \tag{5}$$

$$\ln IITI_{it} = \alpha_0 + \alpha_1' DID_{it} + \alpha_3 \ln ER_{it} + \sum_k \beta_k \ln X_{kit} + \gamma_i + \delta_t + \mu_{it}, \tag{6}$$

where $ER_{it}$ represents the level of energy restructuring of the B&R country i in year t. W signifies a series of control variables, including GDP (constant USD 2010), industrial value added (constant USD 2010), the share of electricity production from oil in total, fixed broadband subscriptions (per 100 people), and kilometres of railway. Other variables in formula (4), formula (5), and formula (6) had the same meaning as the same variable in formula (1). The level of energy restructuring and the control variables in formula (5) were obtained from World Bank Open Data.

The results of the mediation tests are shown in Table 6. The regression results in columns (1) to (3) of Table 6 showed that $\alpha_1$ and $\alpha_1'$ were significantly positive, and $\alpha_1$ was greater than $\alpha_1'$. This supported hypothesis 1. Importantly, $\alpha_2$ and $\alpha_3$ were significantly positive, which indicated that hypothesis 2 was true. Because the control variables in formula (5) were different from that of formula (4) and formula (6), the Sobel test could not be performed. In conclusion, the above tests demonstrated that the BRI not only promotes bilateral EGs IIT

**Table 6. Mechanism test results.**

| Variables | ln *IITI* | ln *ER* | ln *IITI* |
|---|---|---|---|
| | (1) | (2) | (3) |
| *DID* | 0.0306*** | 0.0185** | 0.0297*** |
| | (0.0097) | (0.0090) | (0.0096) |
| *lnER* | | | 0.0727*** |
| | | | (0.0209) |
| *Control* | yes | yes | yes |
| *Country* | yes | yes | yes |
| *Year* | yes | yes | yes |
| Observations | 1910 | 1910 | 1910 |
| R-squared | 0.7806 | 0.9938 | 0.7822 |

Note: Parentheses show standard errors;

***, **, and * indicate statistical significance at 1%, 5%, and 10%, respectively.

directly, but also has an indirect promotional role by boosting the energy restructuring of B&R countries. This validated the positive effect of the BRI on environmental sustainable development.

## Heterogeneity test

**The impact of the BRI on horizontal and vertical bilateral IIT in EGs.** To examine the impact of the BRI on bilateral EGs IIT, we divided the EGs IIT into horizontal and vertical IIT and estimated the effect of the BRI on the two types of EGs IIT. Generally, horizontal IIT refers to intra-industry trade of products that have nearly the same quality but different specifications and appearances, while vertical IIT is intra-industry trade of products that have different quality levels. Greenaway et al. (1995) used the unit value of trade (exports or imports) to express the quality, and defined horizontal IIT as the simultaneous export and import of product where the unit value of exports relative to that of imports is within a range of $\pm\alpha$ [44]. When the relative unit values are outside this range, any IIT is considered to be vertical. $\alpha$ is the dispersion factor, and was equal to 0.15 or 0.25. According to the above criterion, this study calculated the IIT index, horizontal IIT index, and vertical IIT index, respectively. The regression results are shown in column (1) and column (2) of Table 7 when the dispersion factor is equal to 0.15, and column (3) and column (4) of Table 7 display the estimation results when the dispersion factor was equal to 0.25, which allowed us to evaluate the robustness of the estimate results.

We found that the impact of the BRI on vertical IIT was greater than that of horizontal IIT. As depicted in the "Vision and Actions Plan", an integrated upstream and downstream industrial chain in the field of clean and renewable energy will gradually form under BRI promotion, which shows that the key model of renewable energy cooperation is vertical industrial international division of labour, rather than horizontal. As a result, the BRI had a stronger promotion effect on vertical EGs IIT.

**The impact of the BRI on bilateral IIT in different EGs.** This study divided the 54 HS 6-digit code EGs into five groups according to their environmental protection functions, and matched them with the SITC Rev.4 3-digit code. The description of the environmental protection functions of each group and the matching results are shown in Table 8.

**Table 7. Heterogeneity test of horizontal and vertical EGs IIT.**

| Variables | $\alpha$ = 0.15 | | $\alpha$ = 0.25 | |
| --- | --- | --- | --- | --- |
| | Horizontal IIT | Vertical IIT | Horizontal IIT | Vertical IIT |
| | (1) | (2) | (3) | (4) |
| *DID* | 0.0240** | 0.0272*** | 0.0252** | 0.0287*** |
| | (0.0099) | (0.0089) | (0.0100) | (0.0099) |
| *Control* | yes | yes | yes | yes |
| *Country* | yes | yes | yes | yes |
| *Year* | yes | yes | yes | yes |
| Observations | 1910 | 1910 | 1910 | 1910 |
| R-squared | 0.7739 | 0.3648 | 0.7710 | 0.3967 |

Note: Parentheses show standard errors;

***, **, and * indicate statistical significance at 1%, 5%, and 10%, respectively.

To examine the heterogeneous effect of the BRI on different EGs IIT, regression estimates of 14 categories of EGs under the SITC 3-digit code are shown in Table 9. As seen in Table 9, every EGs experienced a positive impact of the BRI except for the EGs with the codes of 635, 711, 776, and 741. The BRI had a significant and positive relationship with the EGs whose codes were 714, 716, 718, 771, and 874. From a functional point of view, the BRI significantly promoted the IIT of EGs that are used to exploit and utilize renewable energy resources and to monitor the environment and pollution. Among them, EGs used for exploiting and utilizing renewable energy resources (718) saw the highest promotion effect of the BRI, which is in line with the conclusions of the mechanism test. Therefore, the above estimates also indicated that the BRI has positive environmental impacts.

**The impact of the BRI on China's EGs IIT with different income countries.** Generally, high-income or middle-income countries have more developed EGs industries because the residents in these countries create a higher demand for a high-quality living environment. The World Bank divided all countries into four classes according to their income level (per capital GDP): low-income countries, lower-middle-income countries, upper-middle-income

**Table 8. Classification of EGs and description of environmental protection functions.**

| No. | SITC code* (HS code* in parentheses) | Main environmental protection functions |
| --- | --- | --- |
| 1 | **635**(441872) | a. Saving resources, such as water |
| 2 | **711**(840290, 840410, 840420, 840490) | b. Reducing pollution by using biomass fuel |
| 3 | [c]**712**(840690); **714**(841182, 841199); **716**(850164, 850231, 850239, 850300); **718**(841290); **741**(841919, 841990); **771**(850490); **776** (854140); **871**(901380, 901390) | c. Exploiting and utilizing renewable energy resources |
| 4 | [c]**728**(847420, 847982, 847989, 847990); **741**(841780, 841790, 841939, 841960, 841989, 851410, 851420, 851430, 851490); **743**(842121, 842129, 842139, 842199); **778**(854390) | d. Used for the treatment of waste, pollutants, etc. |
| 5 | [c]**874**(901580, 902610, 902620, 902680, 902690, 902710, 902720, 902730, 902750, 902780, 902790, 903149, 903180, 903190, 903289, 903290, 903300) | e. Used for monitoring the environment and pollution |

* HS code refers to the Harmonized Commodity Description and Coding System, generally simplified to "Harmonized System" or simply "HS" [61].

*SITC code refers to the Standard International Trade Classification, generally simplified to "SITC".

**Table 9. Heterogeneity test of the impact of the BRI on different EGs IIT.**

| SITC code | Main environmental protection functions | Estimate resultsEstimate results |
|---|---|---|
| 635 | a. Saving resources, such as water | -0.0083 |
| | | (0.0074) |
| 711 | b. Reducing pollution by using biomass fuel | -0.0014 |
| | | (0.0068) |
| 712 | c. Exploiting and utilizing renewable energy resources | 0.0059 |
| | | (0.0058) |
| 714 | | 0.0171** |
| | | (0.0082)) |
| 716 | | 0.0181** |
| | | (0.0077) |
| 718 | | 0.0286*** |
| | | (0.0092) |
| 771 | | 0.0183** |
| | | (0.0090) |
| 776 | | -0.0075 |
| | | (0.0062) |
| 871 | | 0.0089 |
| | | (0.0077) |
| 728 | d. Dealing with waste, pollutants, etc. | 0.0015 |
| | | (0.0088) |
| 743 | | 0.0105 |
| | | (0.0078) |
| 778 | | 0.0135 |
| | | (0.0097) |
| 874 | e. Monitoring the environment and pollution | 0.0179** |
| | | (0.0084) |
| 741 | c. Exploiting and utilizing renewable energy resources | -0.0102* |
| | d. Dealing with waste, pollutants, etc. | (0.0061) |
| | *Control variables* | yes |
| | *Country* | yes |
| | *Year* | yes |
| | Observations | 1910 |

Note: Parentheses show standard errors;

***, **, and * indicate statistical significance at 1%, 5%, and 10%, respectively.

countries, and high-income countries. This study classified all countries in the sample into lower-middle-income countries (including low-income countries and lower-middle-income countries) and upper-middle-income countries (including upper-middle-income countries and high-income countries) to conduct a heterogeneity test. The regression results of the two groups are shown in column (1) and column (2) of Table 10, respectively.

The results showed that the BRI has a greater impact on China's EGs IIT with lower-middle-income countries than that of upper-middle-income countries. According to the law of diminishing marginal benefit, because the energy restructuring process in upper-middle-income countries is generally deeper than that in lower-middle-income countries, the BRI has less impact on their energy restructuring. However, the process of energy restructuring in lower-middle-income countries is slow, and their EGs industries are less complete, so the BRI has a

**Table 10. Heterogeneity test of different income countries.**

| Variables | Lower-middle | Upper-middle |
|---|---|---|
| | **(1)** | **(2)** |
| *DID* | 0.0317** | 0.0343*** |
| | (0.0135) | (0.0120) |
| *Control* | yes | yes |
| *Country* | yes | yes |
| *Year* | yes | yes |
| Observations | 1210 | 1250 |
| R-squared | 0.7753 | 0.7963 |

Note: Parentheses show standard errors;

***, **, and * indicate statistical significance at 1%, 5%, and 10%, respectively.

greater promotional effect on their energy restructuring and relevant industrial upgrading. As a result, the BRI has contributed more to China's EGs IIT with lower-middle-income countries.

**The impact of the BRI on China's EGs IIT with countries in different geographic locations.** Krugman (1991) revealed that the geographic environment has a close relationship with the economic activity of a country [62]. Based on this argument, it is necessary to classify the B&R countries according to their geographic location and test the impact of the BRI on EGs IIT of B&R countries in different geographic locations. This study divided B&R countries into two groups, namely the Silk Road Economic Belt countries (SREB countries) and the 21st Century Maritime Silk Road countries (MSR countries), to test heterogeneity. From column (1) and column (2) of Table 11, we can see that the impact of the BRI on the EGs IIT of MSR countries was greater than that of SREB countries. The MSR countries usually have coastal ports and other convenient transportation facilities, resulting in developed trade, while the SREB countries are usually landlocked and have a less open environment and more inconvenient transportation, which limits their trade development. Therefore, the EGs IIT of MSR countries with China can more easily be promoted by the BRI than that of SREB countries.

## Summary and discussion

This study used the TWFE Staggered DID to examine the impact of the BRI on bilateral EGs IIT between China and B&R countries. Our research results supported hypotheses 1 and 2. The main research conclusions are discussed as follows.

**Table 11. Heterogeneity test of SREB countries and MSR countries.**

| Variables | SREB | MSR |
|---|---|---|
| | **(1)** | **(2)** |
| *DID* | 0.0319* | 0.0289*** |
| | (0.0170) | (0.0103) |
| *Control* | yes | yes |
| *Country* | yes | yes |
| *Year* | yes | yes |
| Observations | 850 | 1620 |
| R-squared | 0.8136 | 0.7816 |

Note: Parentheses show standard errors;

***, **, and * indicate statistical significance at 1%, 5%, and 10%, respectively.

Firstly, the BRI can directly promote China's EGs IIT with B&R countries, and also has an indirect promotional impact by stimulating the energy restructuring of B&R countries. This indicates that the BRI can contribute to improving environment conditions and reducing the carbon emissions in B&R countries, which will be conducive to the sustainable development of B&R countries.

Secondly, the BRI has a greater promotional effect on vertical EGs IIT than horizontal EGs IIT. It's probably because integrated upstream and downstream industrial chain is under forming between China and B&R countries, which indicates the cooperation mode belongs to intra-industry division of labour rather than inter-industry division of labour.

Thirdly, the BRI significantly promotes the IIT of EGs that are used to exploit and utilize renewable energy resources and to monitor the environment and pollution. This validates the conclusion of the mechanism test and also reflects the positive role of the BRI in protecting the environment and reducing carbon emissions.

Fourthly, based on the law of diminishing marginal benefit, because the energy restructuring in upper-middle-income countries is deeper, the BRI has contributed less to their energy restructuring, meaning that the BRI has had a greater promotional effect on bilateral EGs IIT between China and lower-middle-income countries.

Fifthly, the BRI has a greater promotional effect on the EGs IIT of MSR countries than that of SREB countries. This is resulted from the reason that the MSR countries usually have better geographical environment (or location) and more convenient transportation facilities than the SREB countries.

## Policy Implications

The results of this research have vital policy implications. First, considering the environmental impacts of the BRI, the relevant government departments of China and B&R countries should strengthen their policy coordination and take action, especially in boosting the energy restructuring of B&R countries and promoting the development of EGs IIT. Other countries will then be able to understand and support the green B&R construction and green development of the B&R, confirming that it is in favour of environmental protection and climate governance, thereby contributing to the 2030 Agenda for Sustainable Development among B&R countries.

Second, since the positive impact of the BRI on vertical EGs IIT was greater than the impact on horizontal EGs IIT, relevant departments need to analyse the models of intra-industry trade between China and each B&R country. Then they can take differentiated measures to promote bilateral EGs IIT accordingly.

Third, the IIT of the EGs that are used to exploit and utilize renewable energy resources and to monitor the environment and pollution experienced a greater promotional effect of the BRI, which also indicates the positive impacts of the BRI on the environment and climate. To increase the positive environmental impacts of the BRI and promote global climate governance, the Chinese government should negotiate with B&R countries to expand the scope of importation and exportation of relative EGs.

Fourth, since the BRI has a greater promotional effect on the energy restructuring and EGs IIT of lower-middle-income countries, the BRI could narrow the North–South gap in the field of environmental governance. Therefore, the Chinese government should encourage enterprises to cooperate in relevant EGs fields with B&R countries and provide capital, human resources, and technology support so as to promote the environment and climate governance of B&R countries, thereby making contributions to the 2030 Agenda for Sustainable Development.

Finally, as transportation conditions have an important moderation role in the impact of the BRI on bilateral EGs IIT, it is necessary to vigorously promote the infrastructure

development of B&R countries. China should continue to strengthen infrastructure construction, including transportation, energy, and telecommunication facilities, between China and B&R countries.

## Supporting information

**S1 File.**
(DTA)

**S2 File.**
(DTA)

**S3 File.**
(DTA)

**S1 Code.**
(DO)

**S2 Code.**
(DO)

**S3 Code.**
(DO)

## Author Contributions

**Conceptualization:** Yacheng Zhou, Weidong Huo.

**Data curation:** Changjiang Peng.

**Formal analysis:** Yacheng Zhou, Weidong Huo.

**Investigation:** Yacheng Zhou, Feiyu Liu, Changjiang Peng.

**Methodology:** Yacheng Zhou, Feiyu Liu.

**Project administration:** Feiyu Liu.

**Resources:** Weidong Huo.

**Software:** Yacheng Zhou, Changjiang Peng.

**Supervision:** Weidong Huo.

**Validation:** Weidong Huo.

**Visualization:** Yacheng Zhou, Changjiang Peng.

**Writing – original draft:** Yacheng Zhou, Feiyu Liu, Changjiang Peng.

**Writing – review & editing:** Yacheng Zhou, Feiyu Liu, Weidong Huo.

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
