## [Decision Letter · Decision Letter 0]

31 Oct 2023

PONE-D-23-29967Does the “Belt and Road Initiative” Benefit the Environment? Insight from Analysis of Intra-Industry Trade in Environmental GoodsPLOS ONE

Dear Dr. Liu,

Thank you for submitting your manuscript to PLOS ONE. After careful consideration, we feel that it has merit but does not fully meet PLOS ONE’s publication criteria as it currently stands. Therefore, we invite you to submit a revised version of the manuscript that addresses the points raised during the review process.

We look forward to receiving your revised manuscript.

Kind regards,

Ghaffar Ali, PhD

Academic Editor

PLOS ONE

[The study was supported by Liaoning Revitalization Talents Program (Number: 612

XLYC2002116)]

 [The funders had no role in study design, data collection and analysis, decision to publish, or preparation of the manuscript.]

5. We are unable to open your Supporting Information file [Supporting Information.rar]. Please kindly revise as necessary and re-upload.

Reviewers' comments:

Reviewer's Responses to Questions

**Comments to the Author**

1. Is the manuscript technically sound, and do the data support the conclusions?

Reviewer #1: Yes

Reviewer #2: Yes

2. Has the statistical analysis been performed appropriately and rigorously? 

Reviewer #1: Yes

Reviewer #2: Yes

3. Have the authors made all data underlying the findings in their manuscript fully available?

Reviewer #1: Yes

Reviewer #2: Yes

4. Is the manuscript presented in an intelligible fashion and written in standard English?

Reviewer #1: Yes

Reviewer #2: Yes

5. Review Comments to the Author

Reviewer #1: 1. Overall good abstract but the author should add more scientific things in an abstract.

2. Why author choose this topic? What’s the novelty of your topic?

3. The author should discuss technical and scientific things in an abstract.

4. Where is your conceptual framework? Or Need of the study?

5. The author should improve the quality of words instead of the quantity of words.

6. The author should read the latest research papers and use them in the review of the literature.

7. The author should try to improve the methodology and discuss all the technical things in the methodology section.

8. How to verify results? What is the reliability of data and results?

9. The author should add a discussion section as well after the empirical results. And try to discuss all the empirical results properly one by one in the discussion section.

10. Try to be concise and to the point as much as possible.

11. I am not satisfied with the author’s citations because of a few mistakes in the citation. You should rewrite your references and try to use the latest literature.

12. Try to explain your points in an appropriate way.

Reviewer #2: The authors worked on the Belt and Road Initiative concerning the analysis of intra-industry trade and environmental goods. The article is acceptable after some minor revisions.

The authors must describe the abbreviations used from the start, like Belt and Road (B&R).

The study Lacks discussion, the authors should add a discussion section.

6. PLOS authors have the option to publish the peer review history of their article (what does this mean?). If published, this will include your full peer review and any attached files.

Reviewer #1: **Yes: **Natasha Murtaza

Reviewer #2: No

---

## [Author Response · Author response to Decision Letter 0]

9 Jan 2024

Dear Editor,

I am writing to submit a revised version of my manuscript entitled "Does the "Belt and Road Initiative" Benefit the Environment? Insight from Analysis of Intra-Industry Trade in Environment Goods" (Manuscript Number: PONE-D-23-29967) for consideration for publication in PLOS ONE. I appreciate the constructive feedback provided by the reviewers and the editorial team during the initial review process.

In response to the reviewers' comments, I have carefully revised the manuscript to address their concerns and improve the overall quality of the paper. The major changes made are summarized in this document titled "Response to Reviewers".

Additionally, I have attached a marked-up copy of the manuscript, highlighting all the changes made to original version. This should facilitate the review process and help the editorial team assess the extent of the revisions.

This manuscript has received the awards (Supported by Liaoning Revitalization Talents Program, the Grant Number is XLYC2002116), so please help us change the statements about the Financial Disclosure and Funding Information if necessary.

I will provide you with the original data and Stata code as a separate file titled "data and code", and they can be made public.

I would like to express my gratitude for the time and effort invested by the reviewers and the editorial team in evaluating my manuscript. Their feedback has been invaluable in enhancing the clarity, rigor, and overall contribution of the work.

I believe that the revisions have substantially strengthened the manuscript, and I hope that the updated version meets the high standards of PLOS ONE. I am confident that the findings presented in this work will make a significant contribution to the field.

Thank you for considering this revised manuscript for publication in PLOS ONE. I look forward to the opportunity to contribute to the scientific community through your esteemed journal.

Sincerely,

Weidong Huo

Professor

School of Finance and Trade, Liaoning University, Shenyang, China

---

## [Decision Letter · Decision Letter 1]

1 Mar 2024

Does the “Belt and Road Initiative” Benefit the Environment? Insight from Analysis of Intra-Industry Trade in Environmental Goods

PONE-D-23-29967R1

Dear Dr. Huo,

We’re pleased to inform you that your manuscript has been judged scientifically suitable for publication and will be formally accepted for publication once it meets all outstanding technical requirements.

Kind regards,

Ghaffar Ali, PhD

Academic Editor

PLOS ONE

Additional Editor Comments (optional):

Reviewers' comments:

Reviewer's Responses to Questions

**Comments to the Author**

1. If the authors have adequately addressed your comments raised in a previous round of review and you feel that this manuscript is now acceptable for publication, you may indicate that here to bypass the “Comments to the Author” section, enter your conflict of interest statement in the “Confidential to Editor” section, and submit your "Accept" recommendation.

Reviewer #2: All comments have been addressed

2. Is the manuscript technically sound, and do the data support the conclusions?

Reviewer #2: Yes

3. Has the statistical analysis been performed appropriately and rigorously? 

Reviewer #2: Yes

4. Have the authors made all data underlying the findings in their manuscript fully available?

Reviewer #2: Yes

5. Is the manuscript presented in an intelligible fashion and written in standard English?

Reviewer #2: Yes

6. Review Comments to the Author

Reviewer #2: As proposed, the authors have responded to all the suggestions and comments in a satisfactory manner. The article "Does the Belt and Road Initiative Benefit the Environment? Insight from Analysis of Intra-Industry Trade in Environmental Goods" has been significantly improved and recommended for publication.

7. PLOS authors have the option to publish the peer review history of their article (what does this mean?). If published, this will include your full peer review and any attached files.

Reviewer #2: No

---

## [Editor Report · Acceptance letter]

19 Mar 2024

PONE-D-23-29967R1 

PLOS ONE

Dear Dr. Huo, 

I'm pleased to inform you that your manuscript has been deemed suitable for publication in PLOS ONE. Congratulations! Your manuscript is now being handed over to our production team.

Kind regards, 

on behalf of

Prof. Ghaffar Ali 

Academic Editor

PLOS ONE